# Coding of stimulus strength via analog calcium signals in Purkinje cell dendrites of awake mice

**Farzaneh Najafi[1†], Andrea Giovannucci[2,3], Samuel S-H Wang[2,3], Javier F Medina[4*]**

[1]Department of Biology, University of Pennsylvania, Philadelphia, United States; [2]Department of Molecular Biology, Princeton University, Princeton, United States; [3]Princeton Neuroscience Institute, Princeton University, Princeton, United States; [4]Department of Psychology, University of Pennsylvania, Philadelphia, United States

**Abstract** The climbing fiber input to Purkinje cells acts as a teaching signal by triggering a massive influx of dendritic calcium that marks the occurrence of instructive stimuli during cerebellar learning. Here, we challenge the view that these calcium spikes are all-or-none and only signal whether the instructive stimulus has occurred, without providing parametric information about its features. We imaged ensembles of Purkinje cell dendrites in awake mice and measured their calcium responses to periocular airpuffs that serve as instructive stimuli during cerebellar-dependent eyeblink conditioning. Information about airpuff duration and pressure was encoded probabilistically across repeated trials, and in two additional signals in single trials: the synchrony of calcium spikes in the Purkinje cell population, and the amplitude of the calcium spikes, which was modulated by a non-climbing fiber pathway. These results indicate that calcium-based teaching signals in Purkinje cells contain analog information that encodes the strength of instructive stimuli trial-by-trial.

**\*For correspondence:** jmed@
psych.upenn.edu

**Present address:** †Cold Spring
Harbor Laboratory, Cold Spring
Harbor, United States

**Competing interests:** The
authors declare that no
competing interests exist.

**Reviewing editor**: Michael
Häusser, University College
London, United Kingdom

## Introduction

The climbing fiber (CF) input to Purkinje cells (PCs) plays a key role in theories of cerebellar learning (*Houk et al., 1996*; *Ito, 2000*) by providing a teaching signal that sounds the alarm when an unexpected sensory event is encountered (*Simpson et al., 1996*; *De Zeeuw et al., 1998*; *Ito, 2013*). Support for this hypothesis comes from studies of Pavlovian eyeblink conditioning (*Medina et al., 2000*), a simple associative task in which subjects learn to blink to an initially neutral cue if it is repeatedly paired with a blink-eliciting instructive stimulus, such as a periocular airpuff. Previous work has shown that some CFs fire a burst of spikes when the unexpected periocular airpuff is delivered (*Sears and Steinmetz, 1991*; *Nicholson and Freeman, 2003*), and that this signal is sufficient for conditioning (*Mauk et al., 1986*; *Thompson et al., 1998*). All of these studies indicate that CFs are an important source of instructive signals to the cerebellum. However, we do not understand how CF signals encode even the most basic stimulus features such as the strength of the periocular airpuff, which has a major impact on the magnitude and the rate of learning (*Spence, 1953*; *Smith, 1968*).

CFs have peculiar physiological properties that have inspired a number of hypotheses about the underlying neural code (*De Zeeuw et al., 2011*; *Najafi and Medina, 2013*). CFs fire bursts spontaneously at ~1 Hz (*Thach, 1968*), a low rate that is barely modulated during sensory stimulation (*Simpson et al., 1996*). Indeed, CFs are often described as binary because they respond by either not firing at all or firing a single burst regardless of how strongly they are stimulated (*Crill, 1970*; *Gibson et al., 2004*). Another peculiarity is that in the adult cerebellum, each PC is innervated by only one CF (*Simpson et al., 1996*). The CF-PC synapse is one of the most powerful and reliable in the brain (*Schmolesky et al., 2002*; *Ohtsuki et al., 2009*): each time the CF fires a burst, it evokes in PCs both

**eLife digest** A region of the brain known as the cerebellum plays a key role in learning how to anticipate an event. For example, if you know that a puff of air is going to be directed at your eye, it's a good idea to close it in advance. However, how much you need to close it depends on how strong that puff of air is. A very strong puff might require closing the eye completely to protect it. In contrast, it is probably better to only partially close the eye if you know a lighter puff of air is coming, so that you can still see.

Extensive research has focused on how neurons in and around the cerebellum work together to achieve this goal. When an event—such as a puff of air—occurs, signals are sent to large neurons in the cerebellum, called Purkinje cells, by 'climbing fibers'. However, climbing fibers were thought to be able to respond in only two ways: either they fire in a single burst to signal that an event has occurred, or they don't fire. It was therefore unclear how the finer details of the event (for example, the strength of the puff of air) are transmitted to the cerebellum.

Najafi et al. imaged the level of calcium in the cerebellum of mice, as this indicates how active the neurons are. When a puff of air was directed at the eyes of the mice, Najafi et al. saw that the size of the response of the Purkinje cells corresponded with how big the puff of air was. Najafi et al. show that the size of this response, which is based mostly on input from the climbing fibers, is also influenced by input from an additional unknown source.

These findings show that Purkinje cells of the cerebellum receive detailed information about the nature of an event, such as a puff of air. What remains to be seen is whether the cerebellum uses this information to learn the correct response, that is how hard to blink to avoid the expected puff.

a burst of sodium spikes in the soma known as a complex spike (*Eccles et al., 1966*; *Thach, 1968*), and a massive calcium-based spike in the dendrite (*Llinás and Sugimori, 1980*).

Based on these observations, it has been suggested that analog information may be encoded probabilistically, in the total number of bursts generated by an individual CF across many repeated presentations of the same stimulus (*Fujita, 1982*; *Kenyon et al., 1998*). Others have pointed out that sensory events synchronize CFs (*Llinás and Sasaki, 1989*; *Lou and Bloedel, 1992*; *Wylie et al., 1995*; *Ozden et al., 2009*; *Schultz et al., 2009*; *Wise et al., 2010*), which raises the possibility that stimulus information may be available in the precise timing of CF inputs (*Schweighofer et al., 2004*; *Van Der Giessen et al., 2008*), or in the level of co-activation in the CF population (*Ghosh et al., 2011*; *Tokuda et al., 2013*). Recent findings suggest an additional possibility: analog information, like the strength of an instructive stimulus, might be encoded post-synaptically by modulating the size of the PC response to individual sensory-driven CF bursts (*Maruta et al., 2007*; *Najafi et al., 2014*; *Yang and Lisberger, 2014*).

We have imaged PC dendrites of awake mice to investigate how CF-triggered calcium events encode information about the strength of a periocular airpuff stimulus. Because each PC receives input from a single CF (*Simpson et al., 1996*), we were able to discriminate the post-synaptic calcium events corresponding to individual pre-synaptic CF bursts and analyze their amplitude, timing and probability. Two-photon imaging also allowed us to analyze ensembles of PC dendrites whose CF inputs were co-activated by the periocular airpuff. Thus our experiments provide a unique opportunity to evaluate a variety of calcium-based codes in PCs, both at the individual dendrite and population levels.

## Results

We used a two-photon microscope to image calcium signals triggered by sensory-driven activation of climbing fiber (CF) inputs to Purkinje cell (PC) dendrites of awake mice. Mice were head-fixed on top of a freely rotating cylindrical treadmill and allowed to locomote in place while we delivered airpuffs of varying pressures and durations to the periocular area. In some experiments we used airpuffs of four different durations (12 experiments, 4 mice, 97 dendrites; durations: 8, 15, 30, 45 ms; pressure: 30 psi); in other experiments we used airpuffs of two different pressures (6 experiments, 3 mice, 39 dendrites; pressures 10, 50 psi; duration: 30 ms). We will refer to these two datasets as duration and pressure data, respectively.

## CF-triggered calcium events in PC dendrites

As in previous reports (*Sullivan et al., 2005*; *Ozden et al., 2008*), PC dendrites in our experiments appeared as parasagittally aligned, tube-like structures (*Figure 1A*), in which large calcium transients (*Figure 1B*, circles) occurred spontaneously and in response to periocular airpuff stimuli (*Figure 1B*, triangles). We have previously shown that these calcium transients are triggered in each individual PC dendrite by activation of its one-and-only climbing fiber (CF) input, which also evokes a complex spike in the PC somata (*Ozden et al., 2008*). Hereafter, we will use the term 'calcium event' to refer to these CF-triggered calcium transients.

A number of features confirmed the CF origin of the calcium events in our experiments. First, they occurred spontaneously at about 1 Hz (0.4–1.4 Hz; median 0.7 Hz; *Figure 1C*), which is similar to the characteristic spontaneous firing rate of CFs reported previously in awake animals (*Thach, 1968*). Second, they had a fast rise (~10 ms *Figure 1D*), as observed for CF-triggered signals in other calcium imaging studies (*Miyakawa et al., 1992*; *Eilers et al., 1995*; *Schmidt et al., 2003*), and a slower decay $t_{1/2}$ of 74 ± 13 ms (mean ± SD), which is in the midrange of previously observed kinetics using synthetic indicators (decay $t_{1/2}$ = 25–170 ms; [*Sullivan et al., 2005*; *Sarkisov and Wang, 2008*; *Ozden et al., 2009*; *Kitamura and Häusser, 2011*]). Third, the probability of observing two spontaneous calcium events at the same time was highest for adjacent PC dendrites, and decreased rapidly as the mediolateral distance between dendrites increased (*Figure 1E*, black). This finding is consistent with previous studies demonstrating the prevalence of synchronous CF input to neighboring PCs in the same parasagittal microzone (*Bell and Kawasaki, 1972*; *Sasaki et al., 1989*; *Ozden et al., 2009*; *Schultz et al., 2009*).

## Location of imaging sites

We imaged 101 sites on the surface of cerebellar cortex in 22 mice, including paravermal locations in lobules V, VI, and more lateral locations in simplex (*Figure 2A,B*). We were less frequently able to image the most medial parts of simplex due to the high density of blood vessels in that area. Consistent with the location of trigeminal CFs reported in previous studies (*Miles and Wiesendanger, 1975a*, *1975b*; *Manni and Petrosini, 2004*), we found that many dendrites in the paravermal regions of lobules V and VI responded to periocular airpuff stimulation with a CF-triggered calcium event (*Figure 2C,D*).

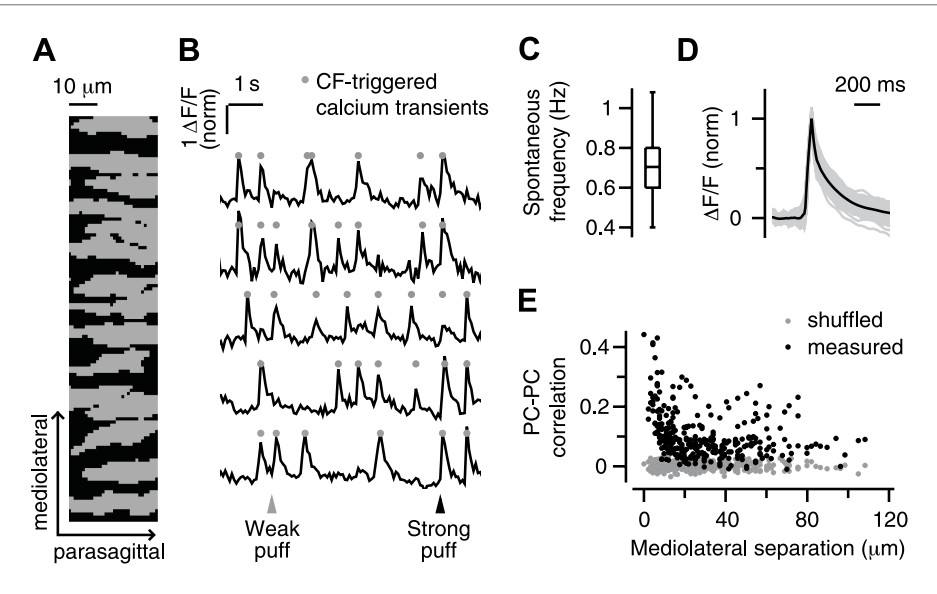

**Figure 1**. Imaging climbing fiber-triggered calcium transients in Purkinje cell dendrites. (**A**) Field of view of an example experiment including 15 dendrites. (**B**) Example fluorescence traces of some of the dendrites in (**A**). Triangles indicate periocular airpuff stimuli of different strengths. Circles mark CF-triggered calcium events. (**C**) Box plot showing frequency of spontaneous calcium events across all dendrites. (**D**) Mean ΔF/F trace of spontaneous calcium events for all dendrites (gray lines; mean: black). (**E**) Pearson correlation coefficient of calcium events in pairs of dendrites as a function of the mediolateral separation (black: real data; gray: shuffled-frame control data).

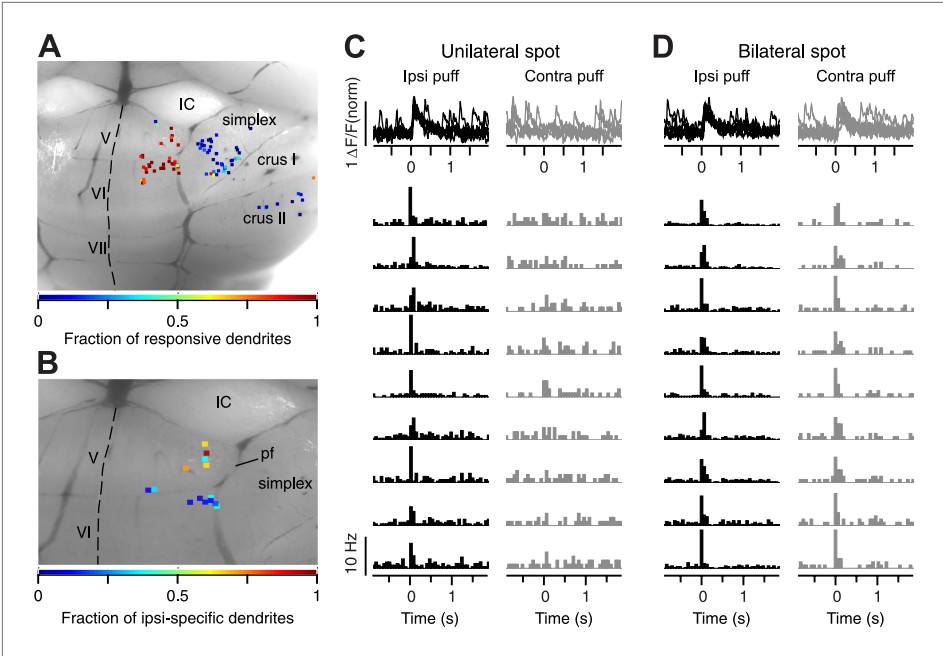

**Figure 2**. Location of imaged spots. (**A**) Dorsal view of an exposed cerebellum showing all imaged spots (colored dots). Colors indicate the fraction of periocular-responsive dendrites in each spot. (**B**) Spots examined for response laterality are shown. Colors indicate the fraction of ipsi-specific dendrites in each spot. IC: Inferior Colliculus. IV–V, VI, VII, Simplex, and CrusI/II: cerebellar lobules. pf: primary fissure. Dashed line: midline. (**C**) Top: example ΔF/F traces from an ipsi-specific spot in response to ipsilateral (black, left) and contralateral (gray, right) airpuff stimuli. Bottom: each row shows PSTH of a dendrite in the example ipsi-specific spot in response to ipsi- and contra-lateral stimuli (left, right, respectively). (**D**) Same as (**C**), but an example bilateral spot is shown.

In contrast, dendrites on the surface of lobule simplex were mostly unresponsive, which is expected given the deep location of periocularly-related CF zones in this lobule (***Hesslow, 1994***; ***Mostofi et al., 2010***; ***Heiney et al., 2014***).

As observed in electrophysiological work (***Hesslow, 1994***; ***Mostofi et al., 2010***), we found two types of periocular PC dendrites that could be classified according to the receptive field properties of their CF inputs: some dendrites responded with a CF-triggered calcium event only after ipsilateral periocular airpuffs (***Figure 2B,C***), while others had bilateral CF receptive fields and responded after ipsi- and contralateral periocular airpuffs (***Figure 2B,D***). In all the analyses presented below, we only evaluated the data for ipsilateral airpuffs. Our results were the same for both types of dendrites.

## CF probability

Previous studies indicate that the likelihood of eliciting a response in an individual CF is proportional to stimulus intensity (***Eccles et al., 1972b***; ***Bosman et al., 2010***). Thus, we first examined if individual CFs encode information about the strength of the periocular airpuff probabilistically, across repeated presentations of the stimulus. The raster plots of all the PC dendrites in our entire duration dataset demonstrate that CF-triggered calcium events occurred more reliably as the duration of the airpuff was increased (***Figure 3A*** bottom; top: average). On average across all the PC dendrites, we found that the probability of calcium events increased gradually with increasing airpuff duration (***Figure 3B***, left; two-way ANOVA: F[496] = 238.5, p < 0.0001; Tukey's HSD, p < 0.01 for all pairwise comparisons), and pressure (***Figure 3B***, right; two-way ANOVA: F[238] = 58.93, p < 0.0001; Tukey's HSD, p < 0.001 for all pairwise comparisons).

The graded increase of calcium event probability with airpuff duration, which is shown averaged across all dendrites in ***Figure 3B***, was evident in 58% of the individual dendrites (***Figure 3C***, top row), in which the probability and the stimulus duration increased with the same rank order. The remaining 42% of the dendrites were relatively unresponsive to all airpuffs below a certain threshold (***Figure 3C***, arrowheads), and responded with similar probability for all durations higher than the threshold (***Figure 3C***,

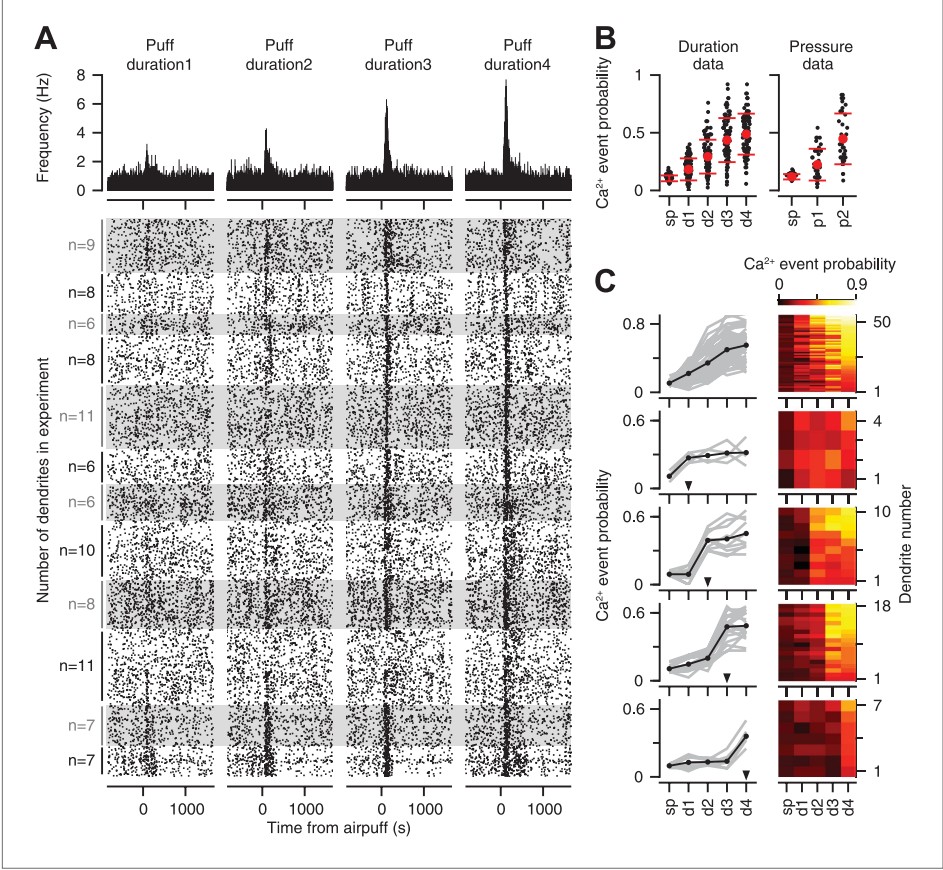

**Figure 3**. Calcium-event probability encodes stimulus strength. (**A**) Bottom: raster plots represent all trials of the duration dataset. Dots indicate calcium events. White and gray shades mark different experiments. For each experiment, all the trials corresponding to an individual dendrite are arranged consecutively. Number of dendrites imaged in each experiment is indicated on the left. Top: PSTHs, corresponding to the raster plots, indicate calcium event frequency at each time point. (**B**) Calcium-event probability for the spontaneous (sp) and airpuff-evoked conditions (d1–d4, p1–p2) (black: individual dendrites; red: mean ± SEM; left: duration data; right: pressure data). (**C**) Top to bottom: five dendrite categories based on how calcium-event probability varies with airpuff duration. For each category, calcium-event probability of individual dendrites (left: gray lines; right: rows of heatmaps) and their average (left, black lines) is shown. Triangles (left): The threshold of airpuff duration for evoking calcium events. Colors (right): calcium-event probability. d1–d4: different airpuff durations. p1–p2: different airpuff pressures.

rows 2–5). These latter CFs may encode duration information about stimulus strength in a binary manner, by providing a signal that tells the post-synaptic PC if the periocular airpuff lasted longer than a certain threshold. In summary, the response probability of individual CFs showed monotonic (i.e., non-decreasing) dependence on stimulus duration, with a dependency that ranged from graded to threshold-like.

## CF latency

In agreement with previous electrophysiological reports of sensory-driven CFs (*Ekerot et al., 1987*; *Kobayashi et al., 1998*), we found that calcium events were evoked in PC dendrites at a wide range of latencies relative to the onset of the airpuff stimulus (onset latency: ~25–150 ms; *Figure 4A,D*). Increasing the duration of the periocular airpuff resulted in progressively more long-latency calcium events (*Figure 4A–C* left; 67.1 ± 1.5, 71.8 ± 1.8, 78.8 ± 1.4, and 80.9 ± 1.4 ms for d1–d4 respectively; two-way ANOVA: F[393] = 11.85, p < 0.0001; Tukey's HSD: p < 0.05 except for d1–d2 comparison and d3–d4 comparison). The temporal jitter, as quantified by median absolute deviation from median latency (*Figure 4C*, right), was also reduced (23.3 ± 1.2, 21.1 ± 1.0, 19.3 ± 0.7, and 18.0 ± 0.8 ms for d1–d4 respectively; two-way ANOVA: F[393] = 5.42, p < 0.01; Tukey's HSD: p < 0.05 for d1–d3, d1–d4,

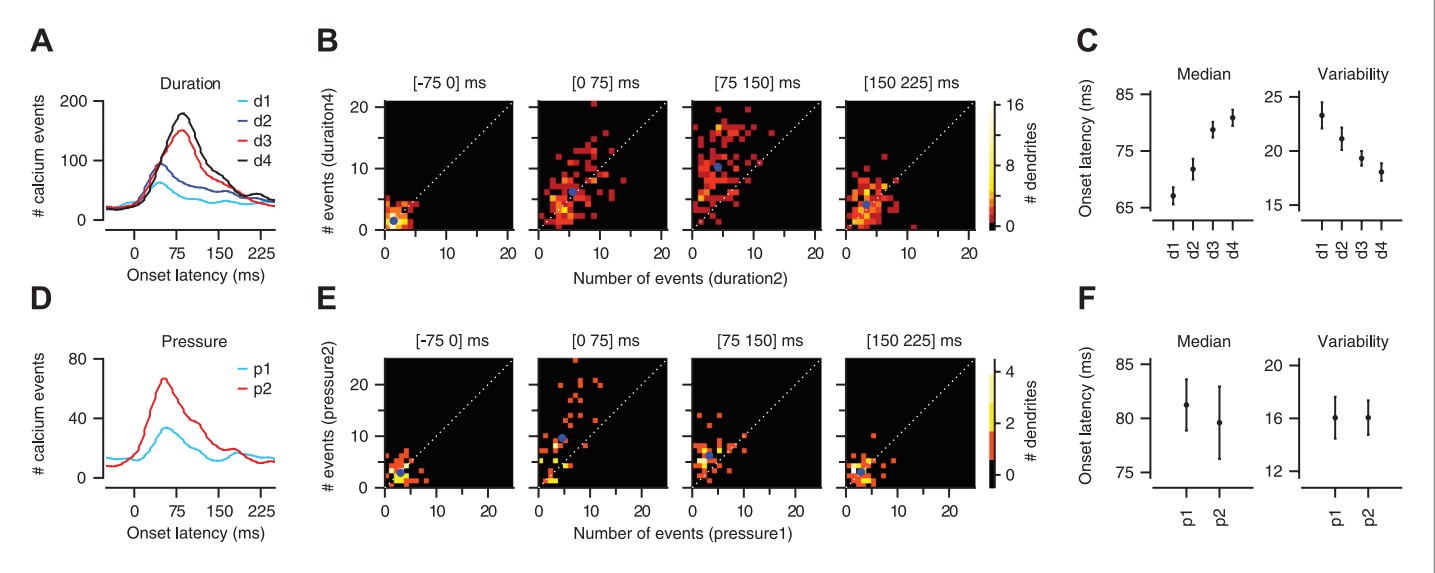

**Figure 4**. Calcium-event latency is modulated by stimulus strength. (**A**) Onset-latency distributions of calcium events for duration data. (**B**) Each panel corresponds to a particular latency interval (indicated in the title) and compares for each dendrite (dots) the number of events evoked by two different durations of airpuff (y-axis: longer duration; x-axis: shorter duration; blue dot: mean; dashed: unity line). (**C**) Median onset latency (left) and variability (median of absolute deviation from median) of onset latency (right). Circles: average across dendrites; Error bars: SEM. (**D–F**) Same as (**A–C**), but for pressure data.

d2–d4 comparisons). The analysis shown in *Figure 4B* confirmed that long-duration airpuffs evoked significantly more calcium events than short-duration airpuffs in the interval 75–150 ms after the periocular stimulation (*Figure 4B*; Kolmogorov–Smirnov test, p < 0.0001) but not in the interval 0–75 ms (*Figure 4B*; Kolmogorov–Smirnov test, p = 0.8). In summary, responses to airpuffs of increased duration are characterized by a higher likelihood of calcium events as the stimulus continues over time.

To test whether CF timing can provide a code for stimulus strength under conditions of constant stimulus duration, we varied the pressure of the airpuff. This led to a modest reduction in onset latency (81.3 ± 2.4 vs 79.6 ± 3.4 ms for p1 and p2 respectively) that was not significant (*Figure 4D–F* left; two-way ANOVA: F[134] = 0.98, p = 0.3), and did not alter temporal jitter (*Figure 4F* right; two-way ANOVA: F[134] = 0.13, p = 0.7). Higher-pressure airpuffs evoked more calcium events throughout the analysis window, both in the 0–75 ms interval (*Figure 4E*; Kolmogorov–Smirnov test, p < 0.01), and in the 75–150 ms interval (*Figure 4E*; Kolmogorov–Smirnov test, p < 0.001). These findings indicate that the onset latency of an individual CF input cannot be used to encode the pressure of the airpuff. However, additional information about stimulus strength may be available in the timing of the CF population (population co-activation). This question is addressed next.

## CF co-activation

Previous work has demonstrated that groups of CFs converging on the same zone of cerebellar cortex become synchronized in response to sensory stimulation (*Lou and Bloedel, 1992*; *Ozden et al., 2009*; *Schultz et al., 2009*; *Wise et al., 2010*; *Ghosh et al., 2011*). We examined if the level of co-activation in the CF population provides information about the strength of the periocular airpuff stimulus. In 16 experiments in which we were able to image at least six PC dendrites simultaneously (*Figure 5*), we found that the number of synchronized calcium events in a 150 ms window after stimulus onset increased in response to airpuffs of longer durations (*Figure 5A,B*, top: two-way ANOVA, F[4394] = 199.75, p < 0.0001; Tukey's HSD, p < 0.05 for all pairwise comparisons), and higher pressures (*Figure 5B*, bottom: two-way ANOVA, F[2209] = 106.56, p < 0.0001; Tukey's HSD, p < 0.0001 for all pairwise comparisons).

Since stronger airpuffs increase the probability of calcium events in individual PC dendrites (*Figure 3*), it is possible that the gradual increase in the number of synchronized calcium events (*Figure 5A,B*) simply reflects an increase in probability in a group of otherwise independent dendrites. To assess this

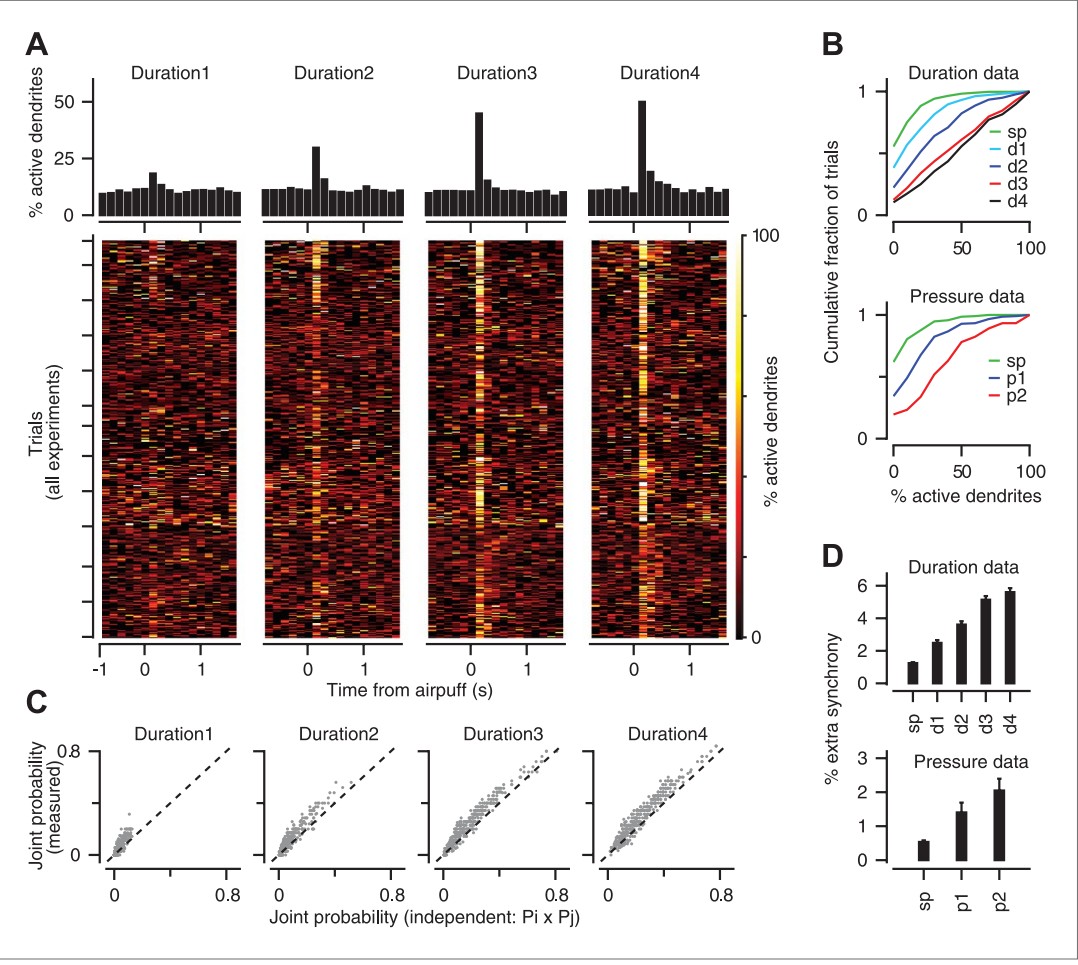

**Figure 5**. Population coding of stimulus strength. (**A**) Bottom: % coactive dendrites at different time points for all trials of the duration data. Colors indicate % coactivation. Top: PSTHs correspond to heatmaps at the bottom and show the average coactivation across all trials at each time point surrounding the stimulus. (**B**) Cumulative distribution of % coactive dendrites across all trials for the spontaneous (sp) and airpuff-evoked conditions (top: duration data; bottom: pressure data). (**C**) Measured and independent joint probabilities are shown for each dendrite pair (gray; dashed: unity line) for different airpuff durations. (**D**) % extra synchrony (measured minus independent joint probability) averaged across all dendrite pairs (error bars: SEM; sp: spontaneous; d1–d4: different airpuff durations. p1–p2: different airpuff pressures).

possibility, we measured the joint probability for every pair of dendrites in each one of our experiments, that is the probability of observing a CF-triggered calcium event in both dendrites within the same 150 ms time window, and compared it to the joint probability expected for independent dendrites ($P_{ij} = P_i \times P_j$, where $P_i$ and $P_j$ represent the calcium-event probability of dendrites i and j in that same time window). We found that joint probability deviated significantly from the independence assumption for all airpuff strengths (*Figure 5C*; two sample *t* test, p < 0.001). We call this deviation 'extra synchrony' because it represents the synchrony beyond that expected solely from the calcium-event probability of two independent dendrites. *Figure 5D* shows that there was a gradual boost in the amount of extra synchrony as the airpuff duration or pressure was increased (Duration data: two-way ANOVA: F[4361] = 113.49, p < 0.0001; Tukey's HSD: p < 0.0001, except for d3–d4 comparison. Pressure data: two-way ANOVA: F[2136] = 10.27, p < 0.0001; Tukey's HSD: p < 0.05, except for p1–p2 comparison). In other words, the degree to which CFs are dependent on each other scales up smoothly with their overall response probability. These results suggest that CF co-activation may be controlled upstream, perhaps in the inferior olive, in a way that provides information about stimulus strength at the level of the PC population.

## Size of CF-triggered calcium events

We have recently shown that the amplitude of CF-triggered calcium events is enhanced during sensory stimulation (*Najafi et al., 2014*). Here, we examined if the magnitude of this sensory-driven enhancement is graded according to periocular airpuff strength. Compared to the average fluorescence traces for spontaneous calcium events (*Figure 6A*, 'sp'), the average fluorescence traces for sensory-driven calcium events revealed a gradual enhancement as the duration or the pressure of the airpuff was increased (*Figure 6A* top; only duration data is displayed). Note that the fluorescence trace of each individual dendrite was normalized to the peak value of its mean spontaneous calcium event ('Materials and methods'). However, calcium events occurred with variable latency after the periocular airpuff, and for this reason the peaks of the average fluorescence traces in *Figure 6A* are substantially lower than '1' (for comparison purposes, the 'sp' trace is plotted with the same temporal jitter as the d1–d4 traces).

To quantify the gradual enhancement in *Figure 6A* (top), we measured the size of each individual calcium event by computing the integral of its fluorescence trace over a 100 ms time window after the peak ('ΔF/F-integral'), and normalizing this value to the average ΔF/F-integral of all the spontaneous calcium events of the corresponding PC dendrite. Note that we only examined the fluorescence traces of individual calcium events and excluded trials in which the periocular stimulation resulted in two or more calcium events separated from each other by less than 100 ms (this occurred in <2% of trials and did not affect our results). This analysis confirmed that periocular airpuffs of longer duration and higher pressure evoked progressively larger calcium events (*Figure 6A*, bottom; Duration data: two-way ANOVA: F[496] = 80.74, p < 0.0001; Tukey's HSD: p < 0.05 except for d2–d3 comparison. Pressure data: two-way ANOVA: F[238] = 62.88, p < 0.0001; Tukey's HSD: p < 0.01 for all pairwise comparisons). Thus, the calcium elevation evoked in PC dendrite after activation of its CF input provides information about the strength of peripheral stimulation.

## Contribution of non-CF signals

What neural mechanisms may contribute to the gradual enhancement of the fluorescence traces in *Figure 6A*? In addition to triggering calcium events in the PC dendrite, sensory stimulation also elicits a smaller calcium response that has a non-CF origin ('non-CF signal'; [*Najafi et al., 2014*]). We found that the rise time (73 ± 8, 114 ± 7, 130 ± 7, 148 ± 7 ms for d1–d4 respectively; two-way ANOVA: F[375] = 12, p < 0.0001; Tukey's HSD: p < 0.05 except for d2–d3 comparison) and the size of this non-CF signal were graded (*Figure 6B*, top), with ΔF/F-integral becoming progressively larger as the stimulus duration or pressure was increased (*Figure 6B* bottom; Duration data: two-way ANOVA: F[496] = 122.61, p < 0.0001; Tukey's HSD: p < 0.0001, except for d3–d4 comparison. Pressure data: two-way ANOVA: F[238] = 71.85, p < 0.0001; Tukey's HSD: p < 0.0001 for all pairwise comparisons). These properties are consistent with the calcium responses driven by activation of parallel fiber inputs to PCs (*Finch and Augustine, 1998*; *Takechi et al., 1998*) which are known to increase progressively with stimulus duration (*Gandolfi et al., 2014*).

For the duration dataset, we had enough trials to make a direct comparison between the size of the non-CF signal and the size of the enhancement of the calcium event, which was obtained by taking the difference between the average fluorescence traces for spontaneous and stimulus-evoked calcium events. For short-duration airpuffs (*Figure 6C*, d1 and d2), the average non-CF signal resembled the average enhancement trace. However, for the longest-duration airpuffs (*Figure 6C*, d4), the non-CF signal was significantly smaller than the enhancement trace. These results are consistent with a model in which calcium events evoked by relatively weak sensory stimulation comprise a constant CF-triggered signal that adds linearly with a non-CF signal graded according to the strength of the stimulus (*Figure 6D*, d1, d2, d3). The supralinear response evoked by very strong airpuffs (*Figure 6D*, d4, arrowhead) could be explained if these stimuli were effective in triggering more spikes in each individual CF burst (*Maruta et al., 2007*).

## Discussion

Our results demonstrate that climbing fibers (CFs) represent information about the strength of a periocular airpuff in a number of ways, both at the individual-CF and population level. Consistent with previous reports of CF activity in awake animals (*Gibson et al., 2004*), we found no evidence for rate coding in our experiments: each presentation of the periocular airpuff resulted in either zero or one calcium event in the Purkinje cell (PC) dendrite, which would indicate that the pre-synaptic CF was activated at most once in virtually all trials (multiple activation in <2% of all trials). In the absence of a

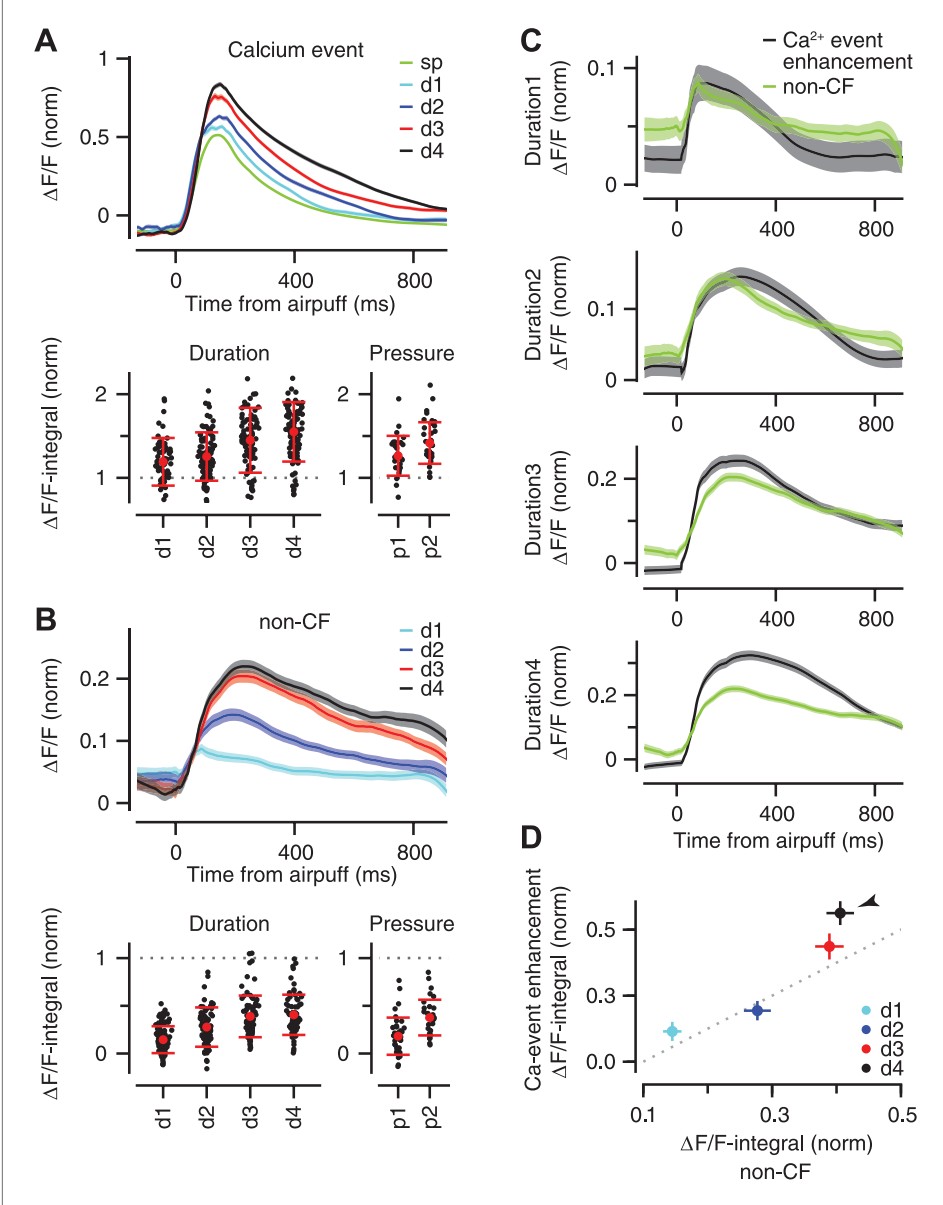

**Figure 6**. Stimulus strength is represented in the size of calcium events and size of non-CF signals. (**A**) Top: mean ΔF/F trace of calcium events across all dendrites for the spontaneous ('sp', green) and airpuff-evoked conditions (d1–d4: different airpuff durations; shades: SEM). Bottom: mean size of calcium events ('ΔF/F-integral') shown for each dendrite (black dots). Left: duration data: Right: pressure data. ΔF/F-integral values are normalized to the mean size of spontaneous events (dashed line). Red: mean ± SEM. (**B**) Same as (**A**), but for the non-CF signal. (**C**) Each panel corresponds to a duration of airpuff, and compares ΔF/F traces of calcium-event enhancement (i.e., evoked minus spontaneous event; black) and non-CF signal (green) in response to that particular airpuff duration (lines: mean across dendrites; shades: SEM). (**D**) Mean size of calcium-event enhancement is compared with mean size of non-CF signal for different airpuff durations (d1–d4; circles: average across dendrites; bars: SEM; dashed: unity line; ΔF/F-integral values are normalized to the mean size of spontaneous events. Arrowhead marks the longest duration airpuff, for which supralinearity is evident).

rate code, our results indicate that analog information about airpuff strength was encoded in: (1) the probability and latency distribution of individual CF inputs across repeated stimulus presentations, (2) the amplitude of the stimulus-evoked calcium event in individual PC dendrites, and (3) the level of co-activation of the CF population.

Below, we evaluate these different codes, and make a number of predictions about the underlying mechanisms and their potential role in regulating the efficacy of instructive signals during cerebellar learning.

## Probabilistic coding in individual PCs

The response of a PC to sensory-driven activation of its CF input was originally described as an 'all-or-nothing' event (*Eccles et al., 1966*). Such a binary response cannot provide analog information about the graded features of an instructive stimulus in a single trial. However, current models of cerebellar function have pointed out that an individual PC could still extract analog information from its CF input by reading out a probabilistic code: by taking into account the reliability (*Fujita, 1982*; *Kenyon et al., 1998*), or the temporal precision (*Schweighofer et al., 2004*; *Van Der Giessen et al., 2008*; *Tokuda et al., 2013*), with which the all-or-nothing CF input is activated across many repeated presentations of the instructive stimulus.

We found that many individual PCs represent information about airpuff duration and pressure in a manner that is consistent with the probabilistic coding hypothesis. Higher-pressure (or longer-lasting) airpuffs evoked CF-triggered calcium events in PC dendrites in a larger fraction of trials than weaker (or shorter) airpuffs. This type of probabilistic coding may be particularly useful for regulating the efficacy of instructive signals during cerebellar tasks that are learned over many training trials. Every time the CF input fires, it triggers a wide range of synaptic changes in the cerebellar cortex (*Schmolesky et al., 2002*; *Ohtsuki et al., 2009*; *Gao et al., 2012*). Thus, the total amount of plasticity induced in the cerebellar cortex during training is likely to depend on the reliability with which CFs are activated by the repeated presentations of the instructive stimulus.

Our observation that the latency of CF-triggered events becomes progressively longer and less variable as a function of stimulus duration may also have implications for cerebellar learning. As is the case for neurons other brain regions (*Dan and Poo, 2004*), plasticity in PCs is strongly influenced by the relative timing of synaptic inputs. For example, parallel fiber (PF) synapses activated just before the CF input get weaker, whereas those activated just after the CF input get stronger (*Piochon et al., 2012*). Therefore, we predict that the total amount of plasticity induced in a PC over the course of multiple training trials will vary as a function of the number of times the instructive stimulus is able to activate the CF within the same small window of time.

## Analog coding on single trials in individual PCs

We have shown that the amplitude of sensory-driven calcium events in a PC dendrite is graded and provides analog information about stimulus strength in single trials, that is more dendritic calcium if CF input fires in response to a strong periocular airpuff, less calcium for weak airpuffs. There are a number of non-mutually exclusive mechanisms that could contribute to this graded regulation of calcium events.

Our results provide considerable support for the possibility that the amplitude of calcium events in PC dendrites may be modulated by sensory-driven activation of a non-CF input. In support of this hypothesis we found that periocular stimulation activated CF and non-CF inputs converging on the same PC dendrite, and that activation of the non-CF input by itself was sufficient to cause a small dendritic calcium response that was graded according to stimulus strength. Although the source of the non-CF signal is unknown, we note that the excitatory parallel fiber (PF) input satisfies two conditions necessary to play such a role: (1) PF and CF inputs with similar receptive fields often converge on the same PC (*Eccles et al., 1972a*; *Eccles, 1973*), and (2) stimulation of PFs generates graded calcium responses in PC dendrites (*Eilers et al., 1995*; *Gandolfi et al., 2014*), that have similar kinetics to the non-CF signal (*Finch and Augustine, 1998*; *Takechi et al., 1998*; *Wang et al., 2000*) and can boost the amplitude of CF-triggered dendritic calcium events (*Wang et al., 2000*).

For the longest duration airpuffs, the amplitude of the non-CF signal was not large enough to fully account for the sensory-driven enhancement of the calcium events. This finding suggests that in addition to the non-CF signal, other mechanisms may also contribute to the modulation of calcium signals in PC dendrites during sensory stimulation. Recent studies have shown that the number of spikes in the presynaptic CF burst varies systematically depending on experimental conditions both in vitro (*Mathy et al., 2009*) and in vivo (*Maruta et al., 2007*; *Bazzigaluppi et al., 2012*), and that having just one extra spike in the CF burst can cause a substantial enhancement of the postsynaptic calcium response (*Mathy et al., 2009*). These observations raise the possibility that the graded modulation of calcium events in our experiments could be driven in part by small increases in the number of spikes of the CF burst (*Najafi and Medina, 2013*; *Yang and Lisberger, 2014*), especially for the longest stimulus durations.

The discovery of graded calcium signals in PC dendrites has important implications for theories of cerebellar learning. Dendritic calcium is responsible for triggering a variety of mechanisms of cellular plasticity that cause short- and long-term modifications in the strength of PC synapses (*Gao et al., 2012*). These plasticity mechanisms are tightly regulated and can be differentially engaged depending on the precise amplitude and duration of the calcium signal in the PC dendrite. For example, small differences in dendritic calcium can influence how much PF synapses will change in vitro (*Coesmans et al., 2004*; *Tanaka et al., 2007*), and can also determine the direction of learning-related changes in the firing of PCs recorded in vivo during eyeblink conditioning (*Rasmussen et al., 2013*). Thus, we predict that the amount and direction of plasticity induced in a PC during cerebellar learning will vary trial-by-trial depending on the dendritic calcium response triggered by each presentation of the instructive stimulus. This type of trial-by-trial regulation of CF-related instructive signals has been recently demonstrated in monkeys learning a cerebellar-dependent eye movement task (*Yang and Lisberger, 2014*).

## Population coding in PC ensembles

Previous studies have reported that CFs converging on the same parasagittal strip of cerebellar cortex fire synchronously to signal the occurrence of an unexpected stimulus (*Llinás and Sasaki, 1989*; *Lou and Bloedel, 1992*; *Ozden et al., 2009*; *Schultz et al., 2009*; *Wise et al., 2010*). Our results confirm and extend these findings in two ways: First, we demonstrated that the level of synchronization is regulated by the strength of the stimulus, that is a larger fraction of the local CF population was activated after strong periocular airpuffs than after weak airpuffs. Second, we found that stimulus-related differences in the level of synchronization are partly driven by a mechanism that boosts synchrony beyond what is expected for independent CFs. We do not know the source of this extra synchrony or the mechanisms that modulate it during sensory stimulation. However, we note that electrical coupling of cells in the inferior olive (IO) plays an important role in synchronizing CF activity (*De Zeeuw et al., 1997*; *Blenkinsop and Lang, 2006*; *Van Der Giessen et al., 2008*), and that the coupling coefficient can be dynamically modulated up and down via stimulus-related activation of synaptic inputs to the IO (*Llinás and Sasaki, 1989*; *Lang, 2002*; *Lefler et al., 2014*; *De Gruijl et al., 2014b*).

Regulation of CF synchrony could be used to control the efficacy of plasticity signals (*De Gruijl et al., 2012*; *Tokuda et al., 2013*). For example, sensory events that activate many CFs simultaneously may be particularly effective for triggering NMDA-dependent plasticity in molecular layer interneurons of the cerebellar cortex (*Duguid and Smart, 2004*), via synchronized spillover of glutamate from multiple CF release sites (*Szapiro and Barbour, 2007*; *Mathews et al., 2012*). Another possibility is that CF synchrony could be used to set the strength of the plasticity signals sent by inhibitory PCs to downstream cells of the deep cerebellar nuclei (DCN; *Otis et al., 2012*), which are the final output of the cerebellum. This hypothesis is consistent with recent work showing that DCN cells exhibit rebound firing immediately after being released from the hyperpolarization caused by CF-driven synchronization of the PC population (*Hoebeek et al., 2010*; *Bengtsson et al., 2011*), and that this rebound can serve as a trigger for plasticity (*Pugh and Raman, 2006*). Release from hyperpolarization may also be sufficient to trigger calcium-based excitable events in dendrites of the DCN (*Schneider et al., 2013*), an effect that may be enhanced by direct monosynaptic excitation of DCN neurons by CF collaterals firing at the same time (*Llinás and Muhlethaler, 1988*).

In addition to playing a role in the modulation of plasticity signals in the cerebellum, sensory-driven regulation of CF synchrony could also have a more direct and immediate effect on motor control (*Lang et al., 1999*; *Kitazawa and Wolpert, 2005*; *Llinás, 2011*). Increased CF synchrony is observed at the onset of movement (*Welsh et al., 1995*; *Ghosh et al., 2011*; *Ozden et al., 2012*), and the level of synchrony has been shown to correlate with the timing and the velocity of spinocerebellar reflexes after sensory perturbations (*De Gruijl et al., 2014a*). In our experiments, a higher level of CF synchrony after a strong periocular airpuff might serve to enhance the reflex response by generating a faster or longer duration eyeblink movement (*Evinger et al., 1991*).

## Conclusions and future directions

CFs are thought to play the role of teachers (*Simpson et al., 1996*; *De Zeeuw et al., 1998*; *Ito, 2013*), providing the instructive signals that trigger the mechanisms of plasticity necessary for cerebellar learning (*Gao et al., 2012*). To be useful, CFs must do more than alert about the occurrence of an unexpected event such as the presentation of a periocular airpuff in the early stages of eyeblink conditioning (*Sears and Steinmetz, 1991*; *Nicholson and Freeman, 2003*); they must also provide analog information

to indicate how unexpected the event was, that is, how much stronger or weaker the airpuff was compared with expectations. Our experiments demonstrate that CFs and non-CF pathways together encode information about stimulus strength in PC dendrites when the presentation of the stimulus cannot be predicted. It will be important to understand whether and how the different calcium-based codes we have uncovered may be modulated during cerebellar learning, as the stimulus becomes predictable.

## Materials and methods

### Animal preparation

Experimental procedures were approved by the Princeton University Institutional Animal Care and Use Committee and performed in accordance with the animal welfare guidelines of the National Institutes of Health. The details of the animal preparation have been described previously (*Najafi et al., 2014*). Briefly, C57BL/6J mice (female, 8–14 weeks) were deeply anesthetized by inhalation of isoflurane (3–5% induction; 0.5–1.5% maintenance). A small area of the cerebellum was exposed (diameter: 3 mm), and a Kwik-Sil plug, pre-molded on a coverslip, was secured over the dura using a two-piece, stainless steel headplate. Throughout the surgery, sterile saline was used to keep the dura wet. Animals' body temperature was monitored and maintained near 37°C. Analgesics (Meloxicam, 5 mg/kg, subcutaneous) were injected. At the end of the surgery, anesthesia was removed and mice were returned to their cage for recovery.

### Imaging CF-triggered calcium events

Two-photon calcium imaging was performed the day after surgery. Mice were anesthetized with isoflurane (3–5% induction; 0.5–1.5% maintenance). Kwik-Sil plug was removed and calcium indicator (Oregon Green 488 BAPTA-1/AM, Invitrogen, Carlsbad, CA) was injected 150–200 μm below dura, by applying brief positive pressure through a glass pipette (*Ozden et al., 2012*). A new Kwik-Sil plug was used, anesthesia was removed, and the animal was transferred and mounted on a cylindrical treadmill integrated with the imaging apparatus. Calcium imaging was performed on awake animals using a custom-built two-photon laser scanning microscope (*Sullivan et al., 2005*). Fluorescence movies (32 × 128-pixel; 64 ms/frame) were recorded using ScanImage software (*Pologruto et al., 2003*). Animals were monitored throughout the experiment with a camera.

PC dendrites were identified from the imaging movies using independent component analysis (*Ozden et al., 2012*). The fluorescence trace (ΔF/F) of each dendrite was computed, frame-by-frame, as (F-Fb)/Fb, where F is the mean fluorescence intensity of the pixels contributing to a dendrite. Fb is the baseline defined as the lowest eighth percentile of the fluorescence values within a 1-s window. Calcium transients were identified as CF-evoked calcium events using a two-step process (*Ozden et al., 2012*): first, the kinetic properties of the ΔF/F signal had to match a template. Second, the peak amplitude of the transient had to be larger than a predefined threshold.

### Periocular stimulation

A pressure injector system (Toohey Spritzer) connected to a 25 gage needle was used to deliver airpuffs to the animal's ipsilateral eye (inter-trial interval: 4 s; 35 trials per airpuff condition per experiment). Two different sets of experiments were performed. In one set, 'duration' experiments, the airpuff pressure was kept the same (30 psi) and four different durations of airpuffs were applied (8, 15, 30, 45 ms). In another set, 'pressure' experiments, the airpuff duration was constant (30 ms), and two different airpuff pressures (10, 50 psi) were applied.

### Data analysis

The ΔF/F traces presented in all figures were normalized. Normalization was done for each dendrite separately, by dividing the ΔF/F trace of the dendrite by the peak value of its mean spontaneous calcium event. Calcium events with a peak in the interval 50–200 ms after the periocular stimulus were considered airpuff-evoked. The onset latency of calcium events (*Figure 4*) was computed manually by inspecting each airpuff-evoked calcium event. Coactivation (*Figure 5*) was computed for each trial separately by dividing the number of dendrites with an airpuff-evoked calcium event, by the total number of responsive dendrites in the field of view. The measured joint probability for a pair of dendrites (*Figure 5C*) was calculated as the fraction of trials in which both dendrites in the pair exhibited airpuff-evoked calcium events. The independent joint probability was computed by multiplying the calcium-event probability of the two dendrites ($P_{ij} = P_i \times P_j$). The d1–d4 traces in *Figure 6A* were computed by averaging the normalized ΔF/F signals of all the individual dendrites in trials with a calcium event.

To provide a fair comparison, the 'sp' trace was computed the same way, after substituting the ΔF/F signal of all the stimulus-evoked calcium events in a given dendrite with the normalized ΔF/F signal corresponding to the average spontaneous calcium event for that dendrite. Thus, the temporal jitter in the onset latency of the individual calcium events comprising the 'sp' trace was the same as in the d1–d4 conditions. The size of individual calcium events (ΔF/F-integral, *Figure 6*) was computed by taking the integral of the normalized ΔF/F signal over the interval [t t + 100 ms], where t is the time point at which the peak of the event occurs. The non-CF signal (*Figure 6B,C*) was analyzed in trials without any calcium events within 50–200 ms of the airpuff stimulus. The size of the non-CF signal was measured in a similar way as the calcium events, by taking the integral of the normalized ΔF/F signal over the interval [t t + 100 ms], where t is a time point selected randomly within 50–200 ms of the stimulus. Error bars in figures indicate SEM. Values in text are mean ± SEM unless otherwise specified. Tests of significance were two-tailed.

## Acknowledgements

We thank A Kloth for help with eyelid acquisition software, I Ozden and D Dombeck for help with imaging in awake mice, and C Arlt and C Wilms for comments on an early version of the manuscript. This work was supported by grants to JM (Searle Scholars Program, NIH R01 MH093727), AG (New Jersey Commission on Brain Injury Research CBIR12FE1031), and SS-HW (NIH R01 NS045193, WM Keck Distinguished Young Investigator, NIH RC1 NS068414).

## Additional information

### Funding

| Funder | Grant reference number | Author |
| --- | --- | --- |
| Searle Scholars Program | | Javier F Medina |
| National Institute of Mental Health | R01 MH093727 | Javier F Medina |
| New Jersey Commission on Brain Injury Research | CBIR12FE1031 | Andrea Giovannucci |
| National Institutes of Health | R01 NS045193 | Samuel S-H Wang |
| National Institutes of Health | RC1 NS068414 | Samuel S-H Wang |
| W.M. Keck Foundation | Distinguished Young Scholars Program | Samuel S-H Wang |

The funders had no role in study design, data collection and interpretation, or the decision to submit the work for publication.

### Author contributions

FN, Designed experiments, Established behavioral and imaging set-up, Acquired data in laboratory of Dr Samuel Wang, Developed analysis, Analyzed data, Wrote first draft of manuscript, Edited manuscript; AG, Designed experiments, Established behavioral and imaging set-up, Developed analysis, Edited manuscript; SS-HW, Designed experiments, Developed analysis, Edited manuscript; JFM, Conception, Designed experiments, Developed analyses, Wrote first draft, Edited manuscript

### Ethics

Animal experimentation: Experimental procedures were approved by the Princeton University Institutional Animal Care and Use Committee (protocol number: 1943-13), and performed in accordance with the animal welfare guidelines of the National Institutes of Health.

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
