## [Decision Letter]

Thank you for sending your work entitled “Coding of stimulus strength via analog calcium signals in Purkinje cell dendrites of awake mice” for consideration at *eLife.* Your article has been favorably evaluated by Eve Marder (Senior editor), a member of the BRE, and 2 reviewers. The following individuals responsible for the peer review of your submission have agreed to reveal their identity: Michael Hausser (Reviewing editor); Tom Otis (peer reviewer).

The Reviewing editor and the reviewers discussed their comments before we reached this decision, and the Reviewing editor has assembled the following comments to help you prepare a revised submission.

The authors have used in vivo two-photon calcium imaging to perform a careful, parametric assessment the population activity in climbing fibers in response to a sensory stimulus (peri-ocular air puffs) of different strength and duration. Given the central role of climbing fibers in cerebellar physiology this is a significant topic of study. The authors are the first to quantify in awake mice how a population of climbing fibers represents a sensory stimulus known to drive learning. They find that these stimuli are represented at a population level in a modular fashion and uncover some tantalizing evidence that the individual climbing fiber inputs to single Purkinje neurons are graded. This latter finding is novel and complements some recent high profile work showing that graded signals represent a population-wide signal relevant to learning (91).

Overall, both reviewers found the work to be thorough and exciting: this is a very well-conducted study with a clear and important message that is of wide interest for a broad audience. Although some of the results could be considered predictable given prior work, the evidence is solid and persuasive. For example, the “extra synchrony” analysis (Figure 5) is particularly elegant and effective. The manuscript should be ready for publication in *eLife* once the following major issues are addressed (editors’ note: minor comments are not shown):

Major comments:

1) It would strengthen the paper if the authors could shed more mechanistic light on the analog calcium signal detailed in Figure 6. For example, it would be particularly compelling if the authors could provide some direct evidence that parallel fiber inputs are responsible for this effect. At the very least, they should provide some speculation about the mechanism.

2) It is very important for the authors to clarify exactly how the normalization procedures in Figure 6 have been conducted. For example, it is unclear how the signals in panel A, top are normalized. Relatedly, in panel B bottom, the dotted line no longer indicates the amplitude of spontaneous calcium events and for the sake of consistency.

3) Non-climbing fiber calcium transients: The authors show that the non-CF transients saturate: d4 does not evoke a larger response than d3 (Figure 6). Yet, they claim that the supra-linear increase in response to d4 as compared to shorter duration stimuli might be expected for a parallel fiber-driven dendritic amplification mechanism. If indeed a parallel fiber-driven mechanism would be active, why is this not visible at d3 – when the non-CF response is identical to that of d4?

4) The authors discuss their work in the light of induction of various forms of plasticity in the cerebellar cortex. This is a wise choice and it is well documented why the learning context is indeed probably highly relevant. However, we cannot exclude the possibility that the current findings are also relevant for short-term effects on motor behavior, e.g. reactions following perturbations (see e.g. CS calcium imaging study of WT and Cx36-/- by De Gruijl et al., 2014 in J Neuroscience). The latter study in fact also shows the mechanisms through microzones (rather than zones). Please consider discussing this issue next to the long-term effects.

---

## [Author Response]

*1) It would strengthen the paper if the authors could shed more mechanistic light on the analog calcium signal detailed in*
Figure 6*. For example, it would be particularly compelling if the authors could provide some direct evidence that parallel fiber inputs are responsible for this effect. At the very least, they should provide some speculation about the mechanism*.

We are very interested in identifying the source of the non-climbing fiber signal in Figure 6. However, this is a question that will require new experiments and is beyond the scope of our current paper. The main point we’d like to make with Figure 6 is that in addition to the traditional climbing fiber pathway, sensory-driven instructive signals can reach Purkinje cells via activation of a climbing fiber-independent pathway that modulates dendritic calcium and provides graded information about stimulus strength.

The reviewers requested that we speculate about the mechanism underlying the non-climbing fiber signal. We suggest in the Discussion (in the section, “Analog coding on single trials in individual Purkinje cells”) that parallel fibers could make a major contribution to the non-climbing-fiber signal. We base this prediction on two known features of the parallel fiber input: (1) parallel fibers and climbing fibers activated by the same stimulus often converge on the same Purkinje cell, and (2) direct stimulation of parallel fibers causes a graded calcium response in the Purkinje cell dendrites that has similar kinetics to the non-climbing fiber signal in our experiments. The relevant paragraph in the Discussion has been highlighted.

*2) It is very important for the authors to clarify exactly how the normalization procedures in*
Figure 6
*have been conducted. For example, it is unclear how the signals in panel A, top are normalized. Relatedly, in panel B bottom, the dotted line no longer indicates the amplitude of spontaneous calcium events and for the sake of consistency*.

We have added a few sentences to clarify the normalization procedures in the top panel of Figure 6. All the *average* fluorescence traces in Figure 6 (top) peak below “1” even though the *individual* fluorescence trace of each dendrite was normalized to the peak of its spontaneous event (i.e. the peak of the average fluorescence trace for spontaneous calcium events is ‘1’ for every dendrite). The reason for this is that the traces in Figure 6 are aligned to the onset of the puff stimulus (not aligned to the peak of the individual calcium events). Because *individual* calcium events occurred with variable latency after the periocular airpuff, the peaks of the *average* fluorescence traces in Figure 6 are substantially lower than “1”.

We have also moved the dotted line in the bottom panel of Figure 6 so that it now indicates the amplitude of spontaneous calcium events (and is consistent with the dotted line in the top bottom panel of Figure 6).

*3) Non-climbing fiber calcium transients: The authors show that the non-CF transients saturate: d4 does not evoke a larger response than d3 (*Figure 6*). Yet, they claim that the supra-linear increase in response to d4 as compared to shorter duration stimuli might be expected for a parallel fiber-driven dendritic amplification mechanism. If indeed a parallel fiber-driven mechanism would be active, why is this not visible at d3 – when the non-CF response is identical to that of d4?*

This raises an interesting point. In Figure 6 (top), the confidence band of the d4 response is above that of d3 for a substantial period. This difference would not be captured by the 100-ms time window we selected for our analyses. We now point this out in the text. Second, supralinearity is by its nature amplification, i.e. what may look like a small change in a particular signal (Figure 6, d3 vs d4) is translated into a much bigger change (Figure 6, d3 vs d4). The difference between 6D and 6B could arise from voltage summation entering a regenerative regime, including a difference in spatial spread. We think there is not an inconsistency; but we agree that there is more to study about this in the future.

Another possibility is that the supra-linearity for d4 is partially driven by mechanisms unrelated to the non-CF response. For example, d3 and d4 could both evoke non-CF signals below threshold for supra-linearity, but d4 could generate climbing fiber responses with a higher number of spikes in a larger fraction of the trials. We have modified the text to address this point and include this possibility.

*4) The authors discuss their work in the light of induction of various forms of plasticity in the cerebellar cortex. This is a wise choice and it is well documented why the learning context is indeed probably highly relevant. However, we cannot exclude the possibility that the current findings are also relevant for short-term effects on motor behavior, e.g. reactions following perturbations (see e.g. CS calcium imaging study of WT and Cx36-/- by De Gruijl et al., 2014 in J Neuroscience). The latter study in fact also shows the mechanisms through microzones (rather than zones). Please consider discussing this issue next to the long-term effects*.

We have added a whole new paragraph in the Discussion, at the end of the section “Population coding in PC ensembles”, to discuss the possibility that the level of climbing fiber synchrony could play a direct role in motor behavior. The reference to De Gruijl et al, 2014 has been added as well.